# Synthesis and Investigation of Novel CHCA-Derived Matrices for Matrix-Assisted Laser Desorption/Ionization Mass Spectrometric Analysis of Lipids

**DOI:** 10.3390/molecules27082565

**Published:** 2022-04-15

**Authors:** Antonio Monopoli, Giovanni Ventura, Andrea Aloia, Fulvio Ciriaco, Angelo Nacci, Tommaso R. I. Cataldi, Cosima D. Calvano

**Affiliations:** 1Dipartimento di Chimica, Università degli Studi di Bari Aldo Moro, Via Orabona 4, 70126 Bari, Italy; giovanni.ventura@uniba.it (G.V.); andrea.aloia@uniba.it (A.A.); fulvio.ciriaco@uniba.it (F.C.); angelo.nacci@uniba.it (A.N.); tommaso.cataldi@uniba.it (T.R.I.C.); 2CNR—Istituto di Chimica dei Composti Organometallici (ICCOM), Bari Section, Università degli Studi di Bari Aldo Moro, Via Orabona 4, 70126 Bari, Italy; 3Centro Interdipartimentale di Ricerca S.M.A.R.T., Università degli Studi di Bari Aldo Moro, Via Orabona 4, 70126 Bari, Italy

**Keywords:** novel MALDI matrix, rationally designed, proton affinity, lipids, phospholipids, milk

## Abstract

A significant area of study and upgrading for increasing sensitivity and general performances of matrix-assisted laser-desorption ionization (MALDI) mass spectrometry (MS) is related to matrix design. Several efforts have been made to address the challenge of low-mass-region interference-free for metabolomics analysis and specifically for lipidomics. To this aim, rationally designed matrices as 4-chloro-α-cyanocinnamic acid (ClCCA) were introduced and reported to provide enhanced analytical performances. We have taken this rational design one step further by developing and optimizing new MALDI matrices with a range of modifications on the CHCA core, involving different functionalities and substituents. Of particular interest was the understanding of the electron-withdrawing (e.g., nitro-) or donating (e.g., methoxy-) effects along with the extent of conjugation on the ionization efficiency. In the present work, ten matrices were designed on a reasonable basis, synthesized, and characterized by NMR and UV spectroscopies and laser desorption ionization. With the assistance of these putative MALDI matrices, samples containing phospholipids (PL), and neutral di-/tri-acylglycerols (DAG, TAG) were investigated using milk, fish, blood, and human plasma extracts. In comparison with CHCA and ClCCA, four of them, viz. [(2E,4E)-2-cyano-5-(4-methoxyphenyl)penta-2,4-dienoic acid] (**1**), [(2E,4E)-2-cyano-5-(4-nitrophenyl)penta-2,4-dienoic acid] (**2**), [(E)-2-cyano-3-(6-methoxynaphthalen-2-yl)acrylic acid] (**6**) and [(E)-2-cyano-3-(naphthalen-2-yl)acrylic acid] (**7**) displayed good to even excellent performances as MALDI matrices in terms of ionization capability, interference-free spectra, S/N ratio, and reproducibility. Especially compound **7** (cyano naphthyl acrylic acid, CNAA) was the election matrix for PL analysis and matrix **2** (cyano nitrophenyl dienoic acid, CNDA) for neutral lipids such as DAG and TAG in positive ion mode.

## 1. Introduction

Matrix-assisted laser desorption ionization (MALDI) mass spectrometry (MS) in the last two decades has been largely applied to the qualitative and quantitative analysis of low molecular weight compounds (LMWC). However, some major drawbacks persist as (i) the variability of signal intensities and resolution between different spots of the same sample, (ii) the presence of abundant interfering matrix-derived peak signals in the low *m*/*z* region, and (iii) the non-homogeneous crystallization that limits the application of spatially resolved imaging [1].

In general, it has been experimented that aromatic compounds derived from benzoic [2,3] or cinnamic acids [4] work well as matrices to investigate proteins, peptides, and lipids, while compounds such as 3-hydroxycholinic acid (3-HPA) [5] and succinic acid [6] are more useful for small polymers and oligonucleotides. From a general literature survey, no universal MALDI matrix exists. From time to time, it is necessary to identify the most suitable one for a given class of analytes, assuring the best performance in terms of reproducibility, resolution, ionization efficiency, and, as wide as possible, dynamic range. Focusing on lipids and metabolites, a vast number of matrices different from the conventional ones were proposed: ionic liquids [7], bases [8], or strong bases as proton sponges were used to improve spot homogeneity [2,9,10]. Electron transfer matrices [11] were employed for chlorophylls [12] and dyes analysis [13], super proton sponges were applied to facilitate the ionization of non-polar compounds [14,15] or neutral sugars [16], and binary matrix combination [17,18,19] were utilized to reduce interfering background signals of LMW analytes [20,21,22,23,24,25,26,27,28]. While most of them were found empirically by testing hundreds of small molecules with high absorption coefficients at the laser irradiation wavelength and thus belong to the ‘first generation’ matrices, others are classified as ‘second generation’ ones since their structure was rationally designed by varying the nature, number, and position of functional groups [29]. In many cases, second generation matrices were demonstrated to work better than archetype ones in terms of crystallization homogeneity, resolution, sensitivity, salt tolerance, high vacuum stability, and laser energy resistance [30]. A well-known rationally designed matrix is 4-chloro-α-cyanocinnamic acid (ClCCA), in which the hydroxyl group of CHCA was replaced by a chlorine atom that demonstrated better performances in ionizing acidic peptides [31]. ClCCA exhibits lower proton affinity (PA) than CHCA, resulting in more efficient analyte protonation, improved signal intensity of acidic analytes, and a more uniform instrumental response of peptides regardless of their amino acid composition [31]. Contrary to other proton transfer matrices that elicited its complete decyanation, ClCCA was found to be very useful for the detection of protonated cyanocobalamin [32,33]. Fulὅp et al. [34] identified 4-phenyl-α-cyanocinnamic acid amide (Ph-CCA-NH_2_) as a novel negative ion matrix for the MALDI-MS analysis and imaging of various lipid classes. Moreover, a boronic analog of CHCA provided a reactive matrix possessing molecular recognition properties toward *vic*-diols, α-hydroxyacids, and aminols [35]. 3,4-dimethoxycinnamic acid was indicated as a new matrix for the in situ detection and imaging of endogenous LMW compounds in rat liver, rat brain, and germinating seed tissue [36]. Putative MALDI matrices for the recognition of anionic sulfatides in complex brain lipid mixtures were evaluated by synthesizing 59 structurally related cinnamic acid derivatives [37,38]. Compared to ESI-MS applied to lipidomics, MALDI-MS provides definite advantages in terms of high sensitivity, high throughput, and ability to implement imaging (MSI) of tissue sections. Further, high salt contents are tolerated, which opens the possibility to add dedicated ones as dopants. Finally, lipid samples and matrices are often readily soluble in organic solvents, so complex extracts can be analyzed without additional and time-consuming purification steps.

Despite the considerable progress in the MALDI-TOF MS hardware, the question of the “optimum” matrix for LMWC has not yet been answered, and there is a continuous search for new MALDI matrices producing a few background ions [39]. In essence, an ideal matrix for lipid analysis would have a high absorption coefficient at the viable laser wavelength and a relatively low matrix background below 1000 Da [40]. With this aim, several closely related CHCA-core compounds were prepared by varying both the degree of unsaturation with a π conjugation extension and the nature of the substituents onto the aromatic ring, introducing electron-withdrawing or donating effects. All MALDI matrices were investigated, and their results were compared with the conventional CHCA and ClCCA ones. As expected, the matrix performances were affected by their PA, laser adsorption, and crystallization properties; this latter behavior remains unpredictable and can be strongly influenced by small changes in the chemical structure. Four suitable matrices with excellent signal-to-noise (S/N) ratios, highly resolved spectra, negligible analyte fragmentation, moderate matrix background, and reproducibility were chosen and applied for the detection of lipids in biological and food samples.

## 2. Results and Discussion

### 2.1. Synthesis and Characterization of Novel Matrices

Targeted matrices were synthesized by following the standard Knoevenagel condensation reaction using cyanoacetic acid and the relative aldehydes [41]. Upon the chemical synthesis and ensuing sample purification by repeated reprecipitations (see experimental section), all prepared compounds were examined by ^1^H-NMR, dissolving them in DMSO-d_6_. The chemical structures of ten putative matrices is reported in Figure 1: **1**, (2E,4E)-2-cyano-5-(4-methoxyphenyl)penta-2,4-dienoic acid; **2**, (2E,4E)-2-cyano-5-(4-nitrophenyl)penta-2,4-dienoic acid; **3**, (2E,4E)-2-cyano-5-(4-(dimethylamino)phenyl)penta-2,4-dienoic acid; **4**, (2E,4E)-2-cyano-5-phenylpenta-2,4-dienoic acid; **5**, (2E,4E)-5-(4-chlorophenyl)-2-cyanopenta-2,4-dienoic acid; **6**, (E)-2-cyano-3-(6-methoxynaphthalen-2-yl)acrylic acid; **7**, (E)-2-cyano-3-(naphthalen-2-yl)acrylic acid; **8**, (2E,2’Z)-3,3’-(1,4-phenylene)bis(2-cyanoacrylic acid); **9**, (Z)-3-(4″-((E)-2-carboxy-2-cyanovinyl)-[1,1′:4′,1″-terphenyl]-4-yl)-2-cyanoacrylic acid; **10**, (E)-2-cyano-3-(quinolin-4-yl)acrylic acid. Note that they are divided in three groups representing a distinguishing core structure of the compound. Table 1 summarizes the available physico-chemical information such as LogP, predicted by ChemDraw Ultra 14.0, molecular formula, and theoretical masses of each compound. While the fifth column in the Table 1 lists the molar extinction coefficients estimated at 345 nm, the last column refers to theoretical proton affinities calculated as explained in the text (*vide infra*).

One of the properties that a MALDI matrix should preserve is the ability to absorb with a high extinction coefficient near the laser wavelength of the instrument. In the present case, lipid analysis was attempted with a neodymium-doped yttrium lithium fluoride (Nd:YLF) laser operating at 345 nm. When other MALDI-MS systems equipped with laser sources emitting between 335 and 355 nm are employed, matrices like ClCCA are not very suitable due to the hypsochromic shift related to the hydroxyl group replacement in the CHCA with a chlorine atom. To extend the conjugation of the archetype matrix CHCA, thus favoring a redshift, some auxochromes were introduced into a series of rationally designed closely related derivatives of cinnamic acid. This aspect was evaluated by assuming that the molar UV–visible absorbance properties in the solid phase and solution are roughly related [38]; each compound was diluted in a mixture of ACN:H_2_O (95:5, *v*:*v*, final concentration 15 µM) and examined by UV–vis spectrophotometry between 250 and 600 nm. The resulting absorption spectra of compounds from **1** through **10** are depicted in Figure 2. From UV spectra analysis, it was clear that the size of the aromatic system and the possible mesomeric and inductive effects of the substituents affect the overall absorption profile of these CHCA derivatives, whereas the substitution of -OH group with basic residues as dimethylamino or quinoline (**3**, **10**) caused an ipsochromic effect, lowering the molar extinction coefficient (ε = 1467 and 3667, respectively) as occurred with ClCCA, where the more extended conjugation or the existence of electron-withdrawing auxochromes in compounds **1**, **2**, **4**, **5**, **8**, **9** gave rise to a hyperchromic shift and a greater absorption was observed at 345 nm, being a valuable factor of MALDI matrices. In general, by replacing the phenyl residue with a naphthyl group (**6**, **7**), extending the vinyl side chain (**1**, **2**), or introducing functional groups with positive mesomeric (+M)-effects such as methoxy (**1**) and nitro (**2**) groups, a useful bathochromic shift of the corresponding absorption profiles was achieved.

Another interesting and important feature of a MALDI matrix is the ability to ionize the investigated compounds as protonated adducts [A+H]^+^ or as radical cations (A^+•^), also exhibiting a low background signal in the low mass range. For this reason, all synthesized matrices were characterized by LDI (laser desorption ionization) by using 1 μL of each solution (50 µmol/µL) deposited on the target plate. In terms of the MALDI process, the easiest explanation for ion formation resulting from laser excitation of absorbent organic material is the single-molecule multiphoton ionization (MPI), which leads to the formation of a matrix radical ion [42]. This process is supposed as a key step of primary ionization by UV MALDI-MS until the analyte protonation takes place [43,44]. The excited state lasts a few nanoseconds, during which one or more photons could be absorbed to lead to the formation of matrix radical cation and a free electron [45,46]. When the radical cations were the most intense peaks, they demonstrated a higher reactivity compared to protonated adducts [M+H]^+^ and so the incidence of primary ionization processes can be advised.

Detailed analysis of the LDI mass spectra in the range 100–1000 *m*/*z* of all matrices in positive ion mode is reported in Figure 3. Except for compounds **2**, **5**, and **10**, where additional peaks were present most likely due to the presence of reaction impurities and/or dimers, trimers, etc., the other compounds generate very few interfering peak signals in the low mass range; very clean spectra were obtained mainly for compounds **1**, **3**, **4**, **6**, and **7**. Four of the newly synthesized matrices preferentially ionize as a radical cation (i.e., **1**, **5**, **6**, **9**) and three as protonated adducts (i.e., **2**, **4**, **10**). While both contributions were observed for matrices **3** and **7**, a sodiated adduct was observed for matrix **8**. It is worth noting that for compound **9** the molecular ion is barely detected. In terms of S/N ratio, compounds **1**, **3**, **4**, **6**, **7**, and **10** generated the highest signals while spectra of compounds **8** and **9** were rather noisy. The results indicated that in the spectra of all matrices, there were two interfering peaks at *m*/*z* 215.125 and 242.284, which were respectively assigned as a sodiated adduct [(C_3_H_6_O)_3_•H_2_O+Na]^+^ of a subunit of polypropylene glycol, which is a ubiquitous polyether plasticizer, and tetrabutylammonium cation [C_16_H_36_N]^+^ as a cross-contaminant, which efficiently ionized in LDI. These two interfering signals were strictly related to LD ionization since they were not detected during NMR or GC-MS analyses of MALDI matrices.

To verify the possibility to be used as MALDI matrices, equimolar mixtures with CHCA were prepared. If the protonated adduct of CHCA appears as a leading peak, then an efficient proton transfer to it occurs. Indeed, the mass spectra of matrices **2**, **5**, **6**, **7**, **8**, and **9** were dominated by the peak signal of [CHCA+H]^+^ at *m*/*z* 190.06, thus suggesting a proficient protonation. Comparable intensities were observed in cases **1**, and **4**, most likely because a mutual proton transfer occurs, while matrices **3** and **10** were not able to ionize CHCA. In the last case of matrix **10**, both the conjugation and substituent onto the phenyl ring strongly affect its protonating capacity. The stark difference was observed with electron-withdrawing substituents such as nitro (**2**) and chloro-cyano moieties (**5**), which stabilize the resulting anion (i.e., the conjugated base) upon CHCA protonation, thus suggesting an efficient proton transfer process due to the remarkable differences in proton affinities. To evaluate these correlations, the theoretical proton affinities (PA) of synthesized matrices (see Table 1) were calculated by density functional theory (DFTB3). Matrices **2**, **9**, and **10** exhibited decreased PA values than that of CHCA, and **4**, **6**, and **7** slightly lower, in the case of matrices **1** and **5**, PA values were higher. Matrices **3** and **8** were characterized by the highest and lowest values, 892.6 kJ/mol and 784.5 kJ/mol, respectively. Perhaps the introduction of the dimethylamino functional group in matrix **3** yielded an undesirable stabilization of the protonated matrix with the consequent increase of PA; further, the elevated bathochromic shift (see UV spectra) resulted in insignificant adsorption at the laser wavelength. The introduction of an asymmetric carboxylic substituent in matrix **8** induced a significant decrease in PA since the negative charge generated after deprotonation is delocalized over the entire conjugated π-electron system.

### 2.2. Application of Potential Matrices to Standard Lipid Mixtures

One of the fundamental requirements for a good MALDI matrix in positive ion mode is its ability to protonate the analyte. The effectiveness of synthesized matrices to ionize/desorb lipids was evaluated using three standards of phospholipids (PL). The following lipid species were chosen: 1,2-dioleoyl-*sn*-glycero-3-phosphoethanolamine (DOPE), monoisotopic mass 743.546 Da, 1,2-palmitoyl-*sn*-glycero-3-phospho-L-serine as a sodium salt (DPPS), 735.505 Da, and 1,2-dimyristoyl-*sn*-glycero-3-phospho-(1′-rac-glycerol) as a sodium salt (DMPG) 666.447 Da, because they occur in significant amounts in dairy products, biological tissues, and cells. In positive ion mode, polar PL should be easily detected as protonated and/or sodiated adducts. PL mixtures at the same concentration were investigated by MALDI-TOF MS as illustrated in Figure 4. Spectra in positive ion mode were dominated by very intense peak signals at *m*/*z* 711.42 and 788.51, corresponding respectively to [DMPG+2Na-H]^+^ and [DOPE+2Na-H]^+^ adducts when matrices **1**, **2**, **4**, **5**, **6**, **7** were employed. Remarkably, the highest signal-to-noise (S/N) ratio was observed with matrix **7** even if a prominent product ion was generated upon a neutral loss (NL, 141 Da, C_2_H_8_NO_4_P) of the phosphorylated polar head. Disodiated adducts of PG and PE were detected by using matrices **3**, **8**, **9**, and **10**, but the resulting spectra were relatively noisy (**3**) or dominated by some interfering matrix peaks (**8**, **9**, **10**). At lower signal intensity, sodiated adducts of DPPS at *m*/*z* 758.49 [DPPS+Na]^+^ and 780.47 [DPPS+2Na-H]^+^, monosodiated adducts of DMPG (*m*/*z* 689.44) and DOPE (*m*/*z* 766.54), sodiated plus potassiated adducts as [DMPG+Na+K-H]^+^ (*m*/*z* 727.39) and [DOPE+Na+K-H]^+^ (*m*/*z* 804.49) were detected. The spectral quality in terms of (S/N) ratio and threshold laser fluences obtained using matrices **1**, **2**, **4**, **5**, **6**, **7** was typically better than that obtained using **3**, **8**, **9**, **10**, and in comparison with CHCA and ClCCA. Since another important property of a high-performing matrix lies in the formation of compact and uniformly distributed crystals, such outcomes might be related to a different crystallization behavior. The optical microscope images at 100× magnification of each matrix blended with the PL mixture are illustrated in Figure 5. A thin, dense, and homogeneous matrix layer was observed on the target covered with matrices **1**, **2**, **4**, **5**, **6**, **7**, while larger crystals, roughly evenly distributed, were visible with matrices **3**, **8**, **10**, which are rather comparable to CHCA and ClCCA. An inhomogeneous big crystal on the spot was observed with matrix **9**. To compare the response of all compounds in the PL analysis, the intensities of a double sodiated adduct [DMPG+2Na-H]^+^ were registered in triplicate on a total of 100 spectra per spot and spot-to-spot. The reproducibility was estimated as relative standard deviation (RSD). The evidence of the higher degree of homogeneity for compounds **1**, **2**, **4**, **5**, **6**, **7** was obtained already during the spectra acquisition; the occurrence of a [DMPG+2Na-H]^+^ signal was always detected in each sampling point, whereas several ‘cold’ points of very low intensity were observed with matrices **3**, **8**, **9**, and **10**. Accordingly, the spot-to-spot reproducibility as RSD was 15%, 18%, 20%, 15%, 22%, 21%, respectively, for matrices **1**, **2**, **4**, **5**, **6**, **7**, and 35%, 45%, and 55% for matrices **10**, **3**, and **8,** following a dried-droplet deposition. Conceivably, each synthesized matrix exhibits an inherent and unpredictable crystallization behavior, which remains a critical parameter not easily predictable of a matrix performance.

As a second set of experiments, triglycerides (TAG) i.e., 1,2,3-tridodecanoylglycerol or trilaurin (LLL, 638.55 Da), trimyristin (MMM, 722.64 Da), tripalmitin (PPP, 806.74 Da), and tristearin (SSS, 890.83 Da) were mixed with each investigated matrix to verify its ability to ionize neutral lipids. An illustrative example of MALDI-TOF mass spectra of an equimolar mixture (1 pmol/µL) of TAG dissolved in chloroform using matrices **1**, **2**, **6**, and **7** is presented in Appendix A. The sodium ion adducts of all four target compounds were easily detectable, and few interfering peaks in the low mass range were observed, except for matrix **6**. According to previous reports [47], no protonated molecules were present in the spectrum while the main peak signals were recognized as sodium adducts at *m*/*z* ions 661.54, 767.61, 829.72, and 913.82 corresponding respectively to [LLL+Na]^+^, [MMM+2Na-H]^+^, [PPP+Na]^+^, and [SSS+Na]^+^. The most significant results in terms of ion abundance and S/N ratios were obtained with matrices **1** and **2**.

A step forward was performed by analyzing a more complex mixture of model compounds, i.e., both polar and neutral lipids known as *EquiSPLASH*^®^ whose composition is reported in Table 2. In the same Table are listed all putative protonated, sodiated, and deprotonated molecules that could be generated by MALDI. Plot A of Figure 6 shows the MALDI-MS spectra in positive ion mode through a heatmap representing the average signal intensities of triplicate analyses of lipid species in a logarithmic scale, normalized to 1.0 in each row. A revealed interesting property is that lipid species examined with matrices **1**, **2**, **6**, and **7** exhibited the highest peak signals. It should be noted, however, that in positive ion mode, PC, LPC, and sphingomyelin species suppress the detection of other lipids [48]. Matrix **2** increased the detectability of TAG and DAG in the presence of PL, thus lowering their suppression effects. MAG, DAG, and TAG, as well as CE species, were preferentially ionized as adducts with alkali metal ions, whereas protonated species were never detected. To attempt a correlation between matrix performances and molar extinction coefficient, proton affinity, and logP, the S/N ratios of 15:0–18:1(d7)PC, 15:0–18:1(d7)PE, and 15:0–18:1(d7)–15:0 TAG, representative respectively of polar, weakly anionic, and neutral lipids, were evaluated. As evidenced in Appendix A, the highest S/N ratios of PC were obtained with matrices **7** and **9**; matrices **1** and **7** were better performing in the case of PE while for TAG the best S/N was obtained by using matrix 2, as already observed in Figure 6A. Matrix **7** seems to demonstrate the best performances for PL analysis, also confirmed by experiments on real samples (*vide infra*). The above data disclosed that with lipid species sensitive to protonation such as PE and PC, pKa and/or PA of the matrix was an important issue affecting the S/N ratio. Indeed, matrices **6** and **1** with similar PA also showed a good S/N ratio for these compounds. Further decreasing the PA values led to detrimental detection effects as demonstrated for some peptides [49]. In the case of TAG, the most relevant parameter seemed the molar extinction coefficient, most likely because these species are preferably ionized as cation adducts so the PA factor is not effective. Therefore, the logP seems to not be a relevant influencing factor, whereas crystallization is an unpredictable critical feature.

Even if CHCA and closed related matrices were used in positive ion mode, the same lipid mixture was investigated in the negative modality to verify their ability to ionize other lipid species in the presence of suppressing PC. From the heatmap of Figure 6B, it appears that compounds **4**, **5**, **10** are good candidates as MALDI matrices for negative polarity detection. The S/N ratio was, however, generally lower compared to positive ion mode, thus confirming that CHCA derivatives are more efficient as proton-donating matrices.

### 2.3. Application of Selected Matrices to Real Samples

As demonstrated above on standard compounds, putative matrices **1**, **2**, **6**, and **7** seemed enough suitable to investigate complex lipid extracts from cow milk, fish, human plasma, and blood by MALDI-MS. Figure 7 reports two positive ion mode MALDI MS spectra of cow milk by using matrices **2** (A) and **7** (B). Both matrices **1** (C) and **6** (D) were used for blood samples and matrix **2** (E) was employed with human plasma. Fish muscle was examined with matrix **7** (F). Each spectrum of plots (A)–(F) of Figure 7 represents a snapshot of the global lipid composition. The lipid species were identified through Online Lipid Calculator (http://www.mslipidomics.info/lipid-calc/, accessed on 2 April 2022) by searching for accurate *m*/*z* ratios setting a tolerance of ±0.05 *m*/*z*. MS/MS data were also acquired to confirm supposed assignments (data not shown). A comparison between spectra of plots A and B of Figure 7, which referred to the same lipid milk extract, highlighted the different behavior of matrices **2** and **7**. As found with standard mixtures (see Figure 6A, Appendix A), matrix **2** allowed a relatively good detection of neutral lipids such as TAG and DGA also when PL occurs in the same extract, thus confirming the mitigation effect of PC suppression in positive modality. In the case of matrix **2**, it is argued that, due to the strong detection of TAG and DAG sodium adducts, not only the PA but also the sodium cation affinity (SCA) plays a pivotal role. Jaskolla et al. [50] rationalize the improved performance of CClCA in detecting the phosphatidylethanolamine chloramines as cation adduct by advoking a lower SCA compared to CHCA (i.e., 177.9 kJ/mol vs. 192.5 kJ/mol). They explained that the stable conformation of both CHCA-Na and CClCA-Na complexes is due to sodium chelation of the carbonyl oxygen of the acid functionality and the nitrogen atom. Being the PA value of compound **2** close to CClCA (802.3 kJ/mol vs. 804.6 kJ/mol), we suggested a similar behavior for SCA due to the occurrence of strong electron withdrawal groups, i.e., nitro and chloro substituent, respectively, which can lead to a lower stabilization of the sodiated complex compared to the other matrices.

According to previous findings [51], the peak signals of TAG 34:0 (*m*/*z* 633.50), TAG 36:0 (*m*/*z* 661.54), and TAG 38:1 (*m*/*z* 687.55) were the most intense. The less abundant ions spaced by 14 Da observed between two consecutive peaks correspond to other sodium adducts of TAG clusters as TAG 35:0 (*m*/*z* 647.52), TAG 32:0 (*m*/*z* 605.47), TAG 38:0 (*m*/*z* 689.57), and TAG 40:1 (*m*/*z* 715.58). Sometimes, less significant signals with +16 Da were observed at *m*/*z* 649.49 ([TAG 34:0+K]^+^), *m*/*z* 677.51 ([TAG 36:0+K]^+^), *m*/*z* 705.52 ([TAG 38:1+K]^+^), being sodium and potassium adducts originated from their occurrence in cow milk or solvent impurities. Other peaks at higher *m*/*z* 827.70 ([TAG 50:2+Na]^+^), 855.74 ([TAG 50:1+Na]^+^), 879.74 ([TAG 52:3+Na]^+^), and 881.76 ([TAG 52:2+Na]^+^) were clearly discernible. Besides TAG species, PC, PE, and sphingomyelin were relatively intense as peak signals in the spectrum of Figure 7A. The stark difference with the spectrum of Figure 7B, which is dominated by PL signals, illustrates how the matrix can affect the results. Peaks corresponding, respectively, to protonated and sodiated PC species were at *m*/*z* 760.59 and 782.57 (PC 34:1), *m*/*z* 758.57 and 780.55 (PC 34:2), *m*/*z* 786.60 and 808.58 (PC 36:2), *m*/*z* 788.62 and 810.60 (PC 36:1) together with lower values due to lysophosphatidylcholines (LPCs) arising from the chemical or enzymatic hydrolysis of the major PL occurring in milk. In fact, peak signals at *m*/*z* 496.34, 518.32, 524.37, and 546.35 were assigned to H^+^ and Na^+^ adducts of LPC (16:0) and LPC (18:0), respectively. As expected from the sample composition, the lipid extracts of cow milk contain various PE species; the most abundant were identified as sodiated adducts of PE 32:4 at *m*/*z* 706.44, PE 34:0 at *m*/*z* 742.54, PE 34:1 at *m*/*z* 740.52, PE 34:4 at *m*/*z* 734.48, PE 36:2 at *m*/*z* 766.54. Moreover, protonated and sodiated adducts of the most abundant sphingomyelins [SM (35:0;1)+H]^+^ at *m*/*z* 703.54 and [SM (35:0;1)+Na]^+^ at *m*/*z* 725.55 were established.

The MALDI MS spectra of blood samples (plots C and D of Figure 7) mainly contain two lipid species, SM and PC, which are generally detectable in lipoproteins [52]. Once again, the major SM was found at *m*/*z* 703.54 as protonated and *m*/*z* 725.55 a sodiated species, whereas the PC moiety exhibited a higher heterogeneity of fatty acyl chains. Considering that PCs are composed mainly of palmitic, stearic, oleic, linoleic substituents and, to a minor extent, arachidonic acyl chain [52], almost all the *m*/*z* values observed in the spectra were easily assigned. Over again, there were peaks associated with protonated adducts of LPC containing one palmitic (*m*/*z* 496.34), stearic (*m*/*z* 524.37), oleic (*m*/*z* 522.36), linoleic (*m*/*z* 520.34) substituent alongside minor LPE containing linoleic acid (*m*/*z* 478.29). Comparing the same sample of human blood, matrix **1** gave rise to the simplest spectrum (Figure 7C) with a good S/N ratio, whereas matrix **6** (Figure 7D) showed several matrix-related interfering peaks. Indeed, as explained, matrix **1** allowed to ionize very well polar lipids as LPC, LPE, PC, and SM with S/N ratios between 100 and 1000 while, for matrix **6**, S/N ratios between 20 and 400 were registered. Moreover, in the *m*/*z* 550–700 range (Figure 7D) some peaks likely correlated to matrix adducts were detected.

As far as the human plasma sample, the use of matrix **2** was found to be the most suitable for ionizing neutral lipids besides the most abundant polar PL (Figure 7E). Indeed, only in this case were the peaks at *m*/*z* 853.73 ([TAG 50:2+Na]^+^), 855.74 ([TAG 50:1+Na]^+^), 879.74 ([TAG 52:3+Na]^+^), and 881.76 ([TAG 52:2+Na]^+^) observed; note that in the plasma samples the lyso-forms are generally more abundant due to their processing and aging [53].

The last application referred to a lipid extract from the *Spaurus aurata* fillet sample, and the MALDI-MS spectrum using matrix **7** is reported as an example in Figure 7F. As demonstrated for milk samples, this matrix allowed the improved detection of polar lipids; in particular, predominant PC and related lyso-forms were observed. PC and LPC at higher molecular weights were detected due to the presence of long-chain acyl chains such as eicosapentaenoic (EPA C20: 5), docosahexaenoic (DHA C22: 5), and docosahexaenoic acids (22:6) in fish.

Conclusively, although all the tested compounds satisfied some specific criteria to act as a useful matrix, such as strong absorption at the emission wavelength of the laser, 4 out 10 compounds (**1**, **2**, **6**, **7**) resulted in enhanced signal-to-noise ratio in real complex samples, which allowed us to detect their representative lipidome.

## 3. Experimental Section

**Materials.** α-cyano-4-hydroxycinnamic acid (CHCA), α-cyano-4-chlorocinnamic acid (ClCCA), 2,5-dihydroxybenzoic acid (DHB), 1,2-dioleoyl-sn-glycero-3-phosphoethanolamine (DOPE) sodium salt, 1,2-dimyristoyl-sn-glycero-3-phospho-rac-(1-glycerol) sodium salt (DMPG), 1,2-dipalmitoyl-sn-glycero-3-phospho-L-serine sodium salt (DPPS), cyanoacetic acid, and aldehydes were purchased from Sigma-Aldrich (St. Louis, MO, USA). The triglycerides trilaurin, trimyristin, tripalmitin, and tristearin were purchased from Supelco (Sigma-Aldrich). Equisplash of 13 deuterated lipid standards at a concentration of 100 µg/mL each was acquired from Avanti Polar Lipids (Alabaster, AL, USA). Dimethyl sulphoxide (DMSO) used as a solvent for NMR analysis was purchased from VARIAN (Palo Alto, CA, USA). Water, acetonitrile (ACN), trifluoroacetic acid (TFA), methanol (MeOH), tetrahydrofuran (THF), and chloroform (CHCl_3_) (Sigma Aldrich, St. Louis, MO, USA) were HPLC grade and were used without further purification. Starting reagents are commercially available (from Aldrich and ABCR) and were used as received.

**Chemical synthesis of matrices.** The synthesis of cyanoacrylic acids **1**–**10** listed in Figure 1 was attempted by a Knoevenagel procedure between cyanoacetic acid and the proper aldehyde. The matrix **9** required an additional synthetic step before Knoevenagel condensation (*vide*
Appendix A). Ammonium acetate was used as a catalyst **[41,49]**. Two grams of cyanoacetic acid (1 equivalent), 0.9 equivalent of each appropriate aldehyde, and 0.15 equivalent of ammonium acetate were refluxed under stirring in enough amounts of toluene (ca. 50 mL). After quantitative separation of by-product water by a Dean–Stark apparatus (ca. 3 h), the reaction mixture was cooled to 50 °C and filtered. The crude product was washed with enough distilled water and purified by repeated reprecipitations. Details on the synthesis procedure (general remarks) alongside NMR spectra (Appendix A) are reported in the Appendix A.

**Matrices characterization**. A solution of each matrix (15 µM) was prepared in ACN:H_2_O (95:5) and the UV spectra were acquired for all samples. For ^1^H-NMR and ^13^C-NMR analysis, the matrices were dissolved in deuterated DMSO (DMSO-d_6_) and a spectrum was registered. Laser desorption ionization analysis was carried out using a solution (50 µmol/mL) of each matrix in ACN:H_2_O (2:1) mixture containing 0.1% TFA which was spotted (1 µL) on the MALDI target plate. For unknown compounds, high-resolution mass spectra were recorded by using Shimadzu LCMS-IT-TOF instrument with the following settings: mass range 50–1000 *m*/*z*, ionization system electrospray ion source in negative ion mode, nebulizer gas nitrogen at 3 bar, dry gas nitrogen at 1.5 L/min and 250 °C, collision gas argon.

Reactions were monitored by GLC and GC-MS techniques by using an Agilent 5890 A gas-chromatograph and an Agilent 6850/MSD 5975C instrument, respectively. Both the instruments were equipped with a capillary column HP-5MS (Agilent, l. 30 m, i.d. 0.25 mm, s.p.t. 0.25 µ). NMR spectra were recorded on a Varian Inova 400 MHz spectrometer. Chemical shift values are given in ppm relative to internal Me4Si.

Identification of the reaction products was accomplished by their preliminary isolation and reprecipitation. Next, the products were identified by comparison of their MS and NMR spectra with those reported in the literature, if possible.

All the data from HRMS (ESI-TOF) and NMR analyses are listed in the Appendix A.

**Reprecipitation and purification of matrices**. After synthesis, the compounds obtained were purified to remove any traces of contaminating salts or unreacted starting reagents that could limit the performance of compounds as MALDI matrices. Furthermore, the presence of impurities in the reaction solvent or salts facilitates the formation of dimers, trimers, tetramers of the matrix itself generating interfering ions [36]. This would render the compounds under investigation useless for the analysis of LMW analytes, such as lipids whose masses are below 1000 *m*/*z*. Different reprecipitation protocols were followed for purification, according to the polarity of every single matrix; if the reprecipitation was not observed, some changes were made to the reference protocols usually employed for the purification of CHCA [54].

**1.** 
**Protocol A**
Dissolution of the matrix (5 mg/mL) in ethanol.Heating the solution at about 40 °C for 2 hFiltration of the solution and subsequent dilution of the filtrate 50:50 (*v*/*v*) with ultra-pure H_2_O and shakingIncubation of the liquid in the refrigerator overnight to allow the precipitation of the solidRemoval of the supernatant and subsequent drying of the purified solidRepetition of the procedure at least in triplicate


**2.** 
**Protocol B**
Dissolution of the matrix (5 mg/mL) in a mixture of acetonitrile: water (70:30)Heating the solution to a temperature close to boilingCooling at room temperature and subsequently in an ice bath to allow the precipitation of the crystalsRemoval of the precipitate by centrifugation or filtration and successive dryingRepetition of the procedure at least in triplicate


Furthermore, the matrices were stored in the dark and at a controlled temperature to avoid their degradation. In particular, the synthesized compounds **1**, **2**, **4**, **5**, **8** were reprecipitated following method A, while compounds **3**, **6**, **7**, **9**, **10** were according to protocol B.

## 4. Sample Preparation

**Standard lipids.** Stock solutions of 10 µg/mL of each standard phospholipid were prepared in MeOH/CHCl_3_ (1:1, *v*:*v*); stock solutions of 10 µg/mL of each standard TAG were prepared in CHCl_3_ while the mixture of 13 deuterated lipid internal standards was diluted at a concentration of 10 µg/mL each in MeOH/CHCl_3_ (1:1, *v*:*v*). Each solution was mixed (1:1 *v*/*v* ratio) with the matrix prepared in methanol at 10 mg/mL concentration; 1 µL of the sample was spotted directly on the target plate and analyzed by MALDI-TOF-MS. The same preparation was followed using CHCA or CClCA as matrices.

**Lipid extracts.** Lipids were extracted from milk, human plasma, and blood following the Bligh and Dyer protocol [55]. Briefly, 3 mL of methanol/chloroform (2:1, *v*/*v*) were added to 800 µL of milk or blood sample and the mixture was left for one hour at room temperature. Then 1 mL of chloroform was added, and the mixture was vortexed for 30 s. Finally, 1 mL of water was added, and the solution was shaken before being centrifuged for 10 min at 2000 rpm. The lower phase containing lipids was dried under nitrogen; the residue was dissolved in 100 µL of MeOH/CHCl_3_ (1:1, *v*:*v*). For fish, the same procedure was applied on 100 mg of solid sample pounded in 800 µL of water.

## 5. Instrumentations

UV-Visible spectra acquisition in the range 250–600 nm was performed using a UV1601 spectrophotometer (Shimadzu Italia S.r.l., Milan, Italy). ^1^H-NMR spectra were recorded with a Varian Unity Plus spectrometer (^1^H 400 MHz). Chemical shifts of ^1^H (δH) in parts per million were determined relative to DMSO-d_6_. A MALDI TOF/TOF 5800 system (AB SCIEX, Darmstadt, Germany) equipped with a neodymium-doped yttrium lithium fluoride (Nd:YLF) laser (345 nm), operating in linear positive ion mode, was used (typical mass accuracy was ≤100 ppm). At least 5000 laser shots were typically accumulated by a random rastering pattern, with a laser pulse rate of 400 Hz and laser fluence of 1 mJ/cm^2^ in the MS mode. The delayed extraction (DE) time was set at 450 ns. DataExplorer 4.0 (AB Sciex) was used to control the acquisitions and to perform the initial elaboration of data; SigmaPlot 11.0 was used to graph final mass spectra. ChemDraw Pro 8.0.3 (CambridgeSoft Corporation, Cambridge, MA, USA) was employed to draw chemical structures.

## 6. Methods for Quantum Computations

SMILES [56] strings were redacted for each neutral and cationic structure, resulting from the protonation at the most reasonable protonation site. Neutral and cation molecules were treated as follows: the SMILES string was converted to an initial tridimensional structure with the help of RDKIT [57]; the resulting atomic coordinates were preoptimized through semiempirical PM7 calculations on OpenMOPAC; a final optimization was afforded using DFT calculations through NWChem [58]. The B3LYP [59] functional and the Def2-TZVP [60] basis set were adopted. The zero-point energy correction was evaluated through vibrational analysis with the same computational parameters. The proton affinities were evaluated as E_neutral_-E_protonated_, i.e., are the opposite of protonation energies, where E is the sum of the optimized molecular structure energy and the vibrational zero-point contribution.

The thermal energy depends on the temperature and has not been taken into account; to give an idea of the magnitude, the kinetic energy of the proton would be 3.71 kJ/mol at 298 K, but it partially vanishes with the difference of the thermal vibrational contributions of the neutral and protonated molecules.

Mirabelli et al. [61] made similar computations on a different set of compounds using all combinations of B3LYP and ωB97X density functionals and of aug-cc-pVTZ and 6-311++G(3df,3pd) basis sets, obtaining results differing by at most 12 kJ/mol. Except for 9-aminoacridine, all results were in agreement with the experimental values. For this reason, our calculations were performed with the B3LYP functional using a single good quality basis set.

## 7. Conclusions

Ten closely related CHCA derivatives were prepared and examined as MALDI matrices of lipid species. As expected, the aromatic system size, along with mesomeric and inductive effects of each substituent, affect the ionization properties along with the size and homogeneity of crystals. Seven matrices exhibited good UV-absorption at the laser wavelength of 345 nm, without the generation of dimer or trimer clusters. Six out of ten matrices, i.e., **1**, **2**, **4**, **5**, **6**, **7**, formed homogeneous co-crystals when mixed with phospholipid mixtures, improving the “spot-to-spot” reproducibility and the signal-to-noise ratios. The successful performance of matrices **1**, **2**, **6**, and **7** in positive ion mode was demonstrated by investigating the lipid profiling of cow milk, plasma, blood, and fish samples. Among them, matrix **7** (cyano napthyl acrylic acid, CNAA) was the election matrix for PL analysis in terms of S/N ratios being even able to compete with an established matrix such as ClCCA. The superior ionization qualities of matrix **2** (cyano nitrophenyl dienoic acid, CNDA), especially in the analysis of neutral lipids such as DAG and TAG, was proved as well.

## Figures and Tables

**Figure 1 molecules-27-02565-f001:**
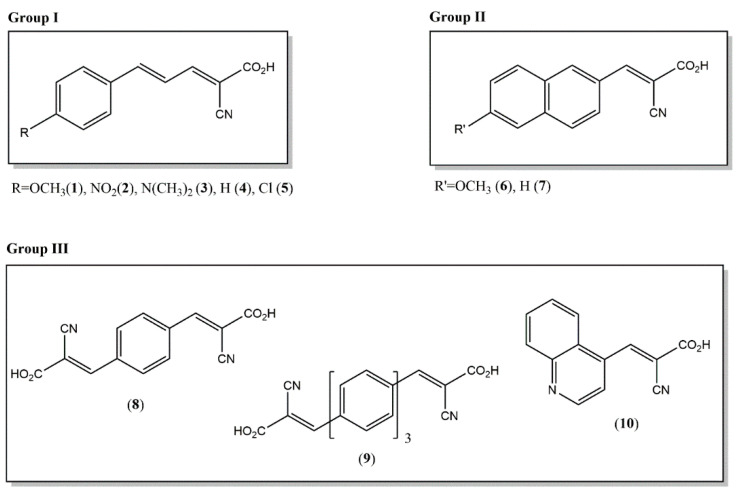
Chemical structures of the synthetized compounds: 2E,4E)-2-cyano-5-(4-methoxyphenyl)penta-2,4-dienoic acid (**1**); (2E,4E)-2-cyano-5-(4-nitrophenyl)penta-2,4-dienoic acid (**2**); (2E,4E)-2-cyano-5-(4-(dimethylamino)phenyl)penta-2,4-dienoic acid (**3**); (2E,4E)-2-cyano-5-phenylpenta-2,4-dienoic acid (**4**); (2E,4E)-5-(4-chlorophenyl)-2-cyanopenta-2,4-dienoic acid (**5**); (E)-2-cyano-3-(6-methoxynaphthalen-2-yl)acrylic acid (**6**); (E)-2-cyano-3-(naphthalen-2-yl)acrylic acid (**7**); (2E,2’E)-3,3’-(1,4-phenylene)bis(2-cyanoacrylic acid) (**8**); (2E,2’E)-3,3’-([1,1’:4’,1’’-terphenyl]-4,4’’-diyl)bis(2-cyanoacrylic acid) (**9**); (E)-2-cyano-3-(quinolin-4-yl)acrylic acid (**10**).

**Figure 2 molecules-27-02565-f002:**
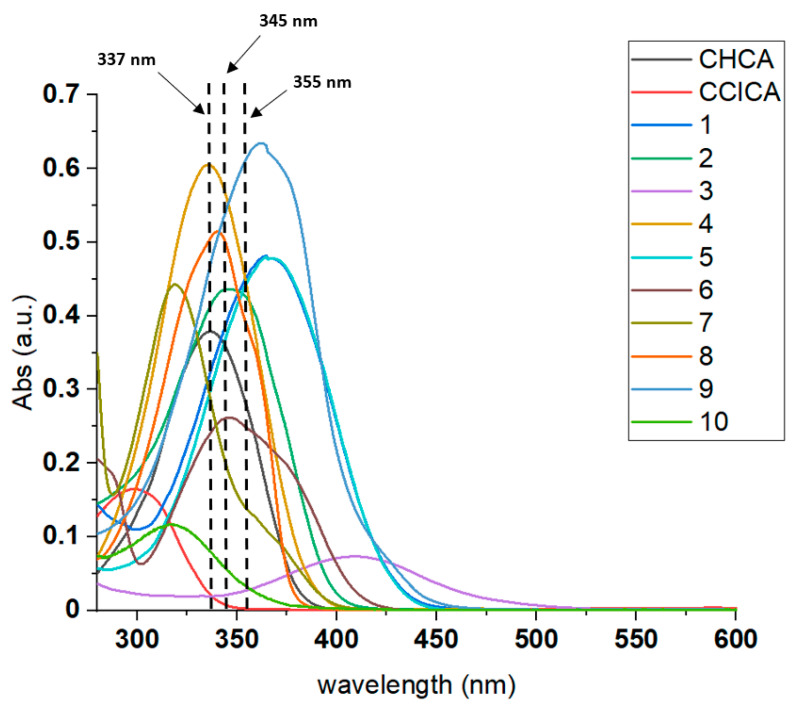
UV absorption spectra of investigated compounds (**1**–**10**) compared with CHCA and ClCCA. The laser wavelengths generally used in MALDI-MS at 337 nm (nitrogen), 345 nm (Nd:YLF), and 355 nm (Nd:YAG) are marked with vertical dotted lines.

**Figure 3 molecules-27-02565-f003:**
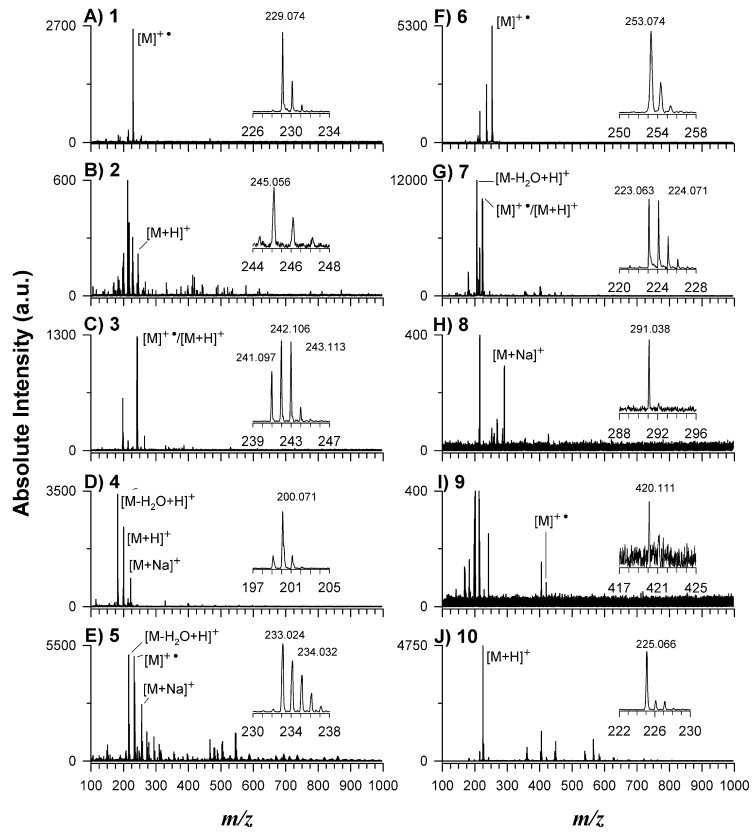
Positive LDI-MS spectra of all investigated compounds from **1** to **10**, illustrated respectively in plots (**A**–**J**). In each spectrum the zoom on the relevant radical, protonated, or sodiated ion adduct is shown.

**Figure 4 molecules-27-02565-f004:**
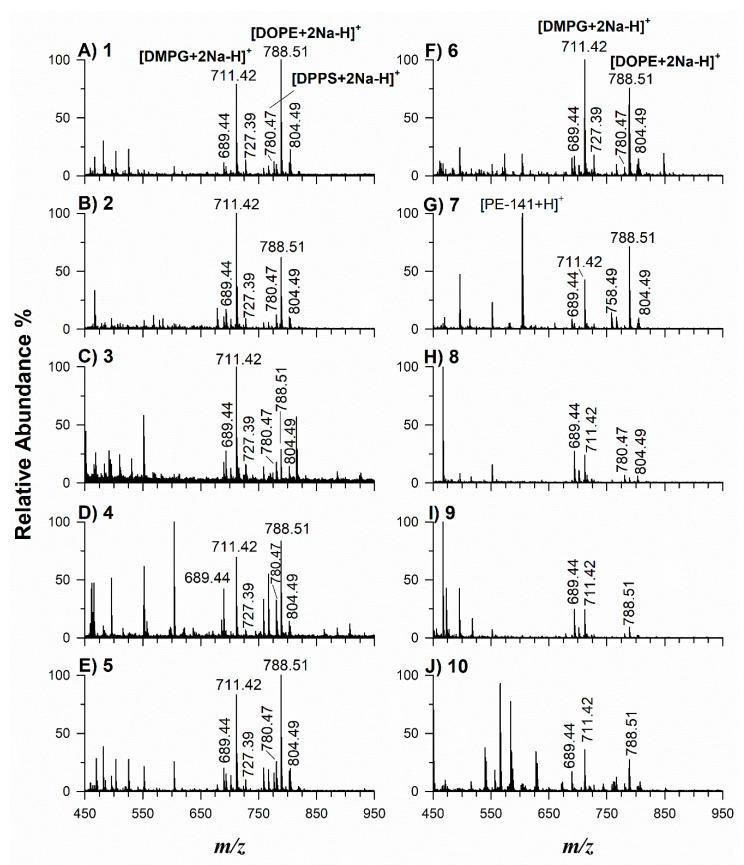
Positive MALDI-ToF MS spectra of the PL mixture composed of sodium salt of 1,2-dioleoyl-sn-glycero-3-phosphoethanolamine (DOPE), 1,2-dimyristoyl-sn-glycero-3-phospho-rac-(1-glycerol) (DMPG), 1,2-dipalmitoyl-sn-glycero-3-phospho-L-serine (DPPS) mixed with each matrix (**A**–**J**).

**Figure 5 molecules-27-02565-f005:**
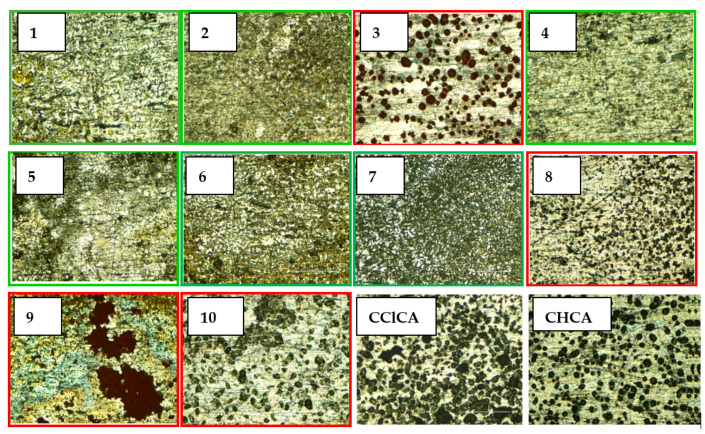
Microscope images of all the matrices (**1**–**10**) after dried-droplet mixing with the phospholipid standard mixture solution. The images of CHCA and ClCCA are reported for comparison.

**Figure 6 molecules-27-02565-f006:**
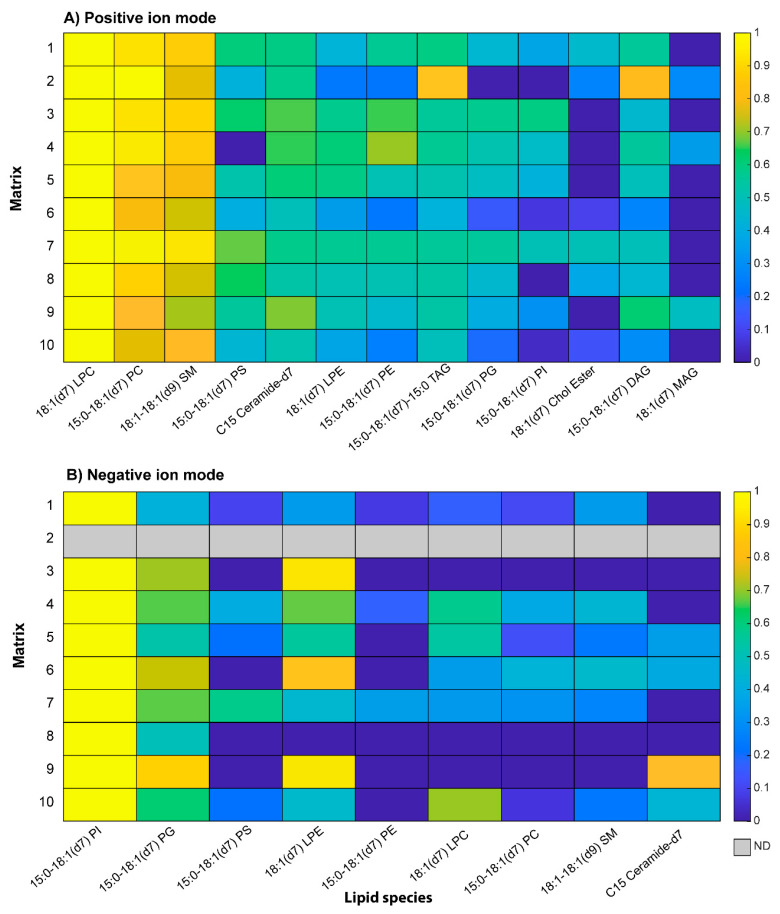
Heatmap plots in logarithmic scale showing the normalized average intensity of each compound mixed with each matrix in positive (**upper panel**) and negative ion mode (**lower panel**). In positive ion mode, the intensities are the sum of sodium and protonated adducts. For TAG, DAG, MAG, PG, and PI, the protonated adduct was not observed regardless of the matrix used, while the sodium adduct was observed for all lipids. In negative ion mode, the deprotonated molecules were considered for PE, LPE, PG, PI, PS, and Cer, while the demethylated species were considered for PC, LPC, and SM. TAG, DAG, MAG, and CholEst do not ionize in negative modality. The color scale index goes from the highest (yellow) to the lowest (blue) intensity.

**Figure 7 molecules-27-02565-f007:**
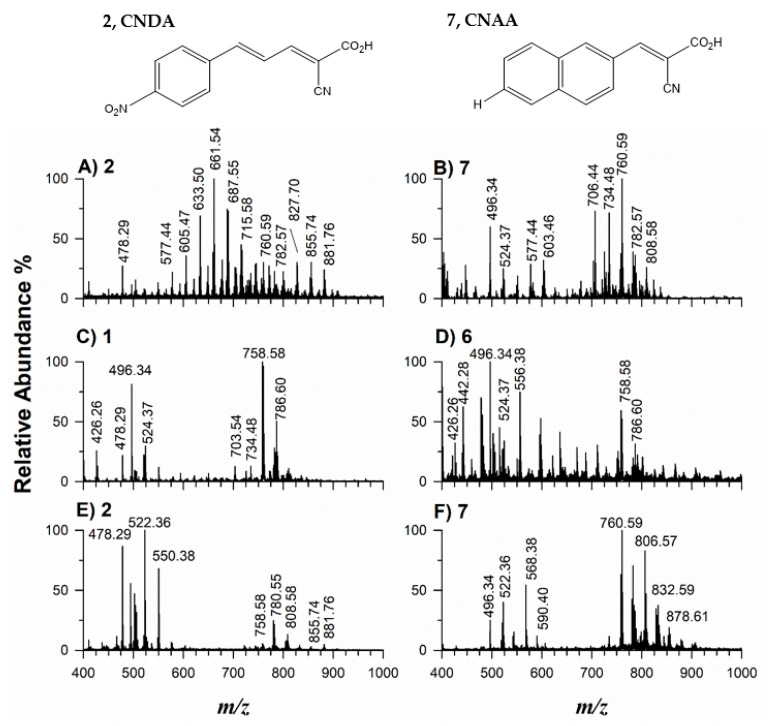
Positive MALDI-ToF MS spectra of the lipid milk extract mixed with matrix **2** (**A**) and **7** (**B**), blood lipids mixed with matrix **1** (**C**) and **6** (**D**), human plasma sample mixed with matrix **2** (**E**), and fish fillet extract blended with matrix **7** (**F**). The structures of best matrices **2** and **7** are reported on the top.

**Table 1 molecules-27-02565-t001:** Summary of the accessible physicochemical information of each compound retrieved from ChemDraw software. The last columns referred to PA theoretically calculated as explained in the text. The molar extinction coefficient was estimated at 345 nm.

Compound n.	LogP	Molecular Formula (M)	Theoretical Mass M (Da)	ε (345 nm)	Proton Affinity (PA, kJ/mol)
**1**	2.22	C_13_H_11_NO_3_	229.0739	26227	872.9
**2**	2.00	C_12_H_8_N_2_O_4_	244.0484	29067	802.3
**3**	2.63	C_14_H_14_N_2_O_2_	242.1055	1467	892.6
**4**	2.34	C_12_H_9_NO_2_	199.0633	29263	838.5
**5**	2.90	C_12_H_8_ClNO_2_	233.0244	27530	874.1
**6**	2.70	C_15_H_11_NO_3_	253.0739	20077	857.3
**7**	2.83	C_14_H_9_NO_2_	223.0633	13067	857.7
**8**	1.62	C_14_H_8_N_2_O_4_	268.0484	65467	784.5
**9**	4.97	C_26_H_16_N_2_O_4_	420.1110	76826	835.2
**10**	1.91	C_13_H_8_N_2_O_2_	224.0586	3667	830.0
**ClCCA**	2.39	C_10_H_6_ClNO_2_	207.0087	23570	804.6
**CHCA**	1.44	C_10_H_7_NO_3_	189.0426	4333	863.2

**Table 2 molecules-27-02565-t002:** List of the mixture of 13 deuterated lipid tested with compounds under investigation. The theoretical values of positive protonated adduct, sodiated adduct, and negative deprotonated adduct are given. For PC, LPC, and SM, the demethylated negative adduct is reported. PC: phosphatidylcholine, LPC: lysophosphatidylcholine, PE: lysophosphatidylethanolamine, LPE: lysophosphatidylethanolamine, PG: phosphatidylglycerol, PI: phosphatidylinositol, PS: phosphatidylserine, TAG: triacylglycerol, DAG: diacylglycerol, SM: sphingomyelin.

Lipid Species	[M+H]^+^ *m*/*z*	[M+Na]^+^ *m*/*z*	[M-H]^−^ *m*/*z*
15:0–18:1(d7)PC	753.61	775.59	736.60 [M-CH_3_-H]^−^
18:1(d7)LPC	529.40	551.38	512.38 [M-CH_3_-H]^−^
15:0–18:1(d7)PE	711.57	733.55	709.55
18:1(d7)LPE	487.53	509.52	485.33
15:0–18:1(d7)PG	742.54	764.52	740.55
15:0–18:1(d7)PI	830.56	852.55	828.56
15:0–18:1(d7)PS	755.55	777.54	753.54
15:0–18:1(d7)–15:0 TAG	812.75	834.74	810.76
15:0–18:1(d7) DAG	588.55	610.54	586.54
18:1(d7) MAG	364.34	386.33	362.33
18:1(d7) Chol Ester	658.65	680.63	656.64
18:1–18:1(d9)SM	738.65	760.63	722.63 [M-CH_3_-H]^−^
C15 ceramide-d7	531.55	553.54	529.533

## Data Availability

Not applicable.

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
