# Peer review of "Synthesis and Investigation of Novel CHCA-Derived Matrices for Matrix-Assisted Laser Desorption/Ionization Mass Spectrometric Analysis of Lipids"

_molecules, 2022, doi:10.3390/molecules27082565_

Round 1

Reviewer 1 Report

The submitted article “Synthesis and investigation of alternative ‘second generation’ matrices for matrix-assisted laser desorption/ionization mass spectrometry analysis of lipids” describes a new suite of rationally designed MALDI matrices for improved analysis of lipids from complex samples. The authors present four new MALDI matrix compounds for improved ionisation, S/N ratios, reproducibility and reduced interference at low m/z. Different recommendations are made for neutral DAGs and TAGs and charged phospholipids.

Although this study is well presented and is informative study, unfortunately, it is my opinion that the experiments are somewhat lacking in key areas of the study, the results are not discussed as fully as they can and should be, and some of the conclusions are not fully supported. I offer the following comments:

Introduction: most of the key literature seems to be presented, although the authors self-cite rather too much.

Page 3 line 114 ‘fifthy’, rather ‘fifth’.

Results: Synthesis and characterization of novel matrices: the authors have not convincingly characterised the synthesised compounds. Table 1 presents LogP predicted by ChemDraw,  theoretically calculated proton affinities and an estimated molar extinction coefficient (based on what exactly?). The only characterisation that seems to be presented is proton NMR, this is not enough and in my opinion is not acceptable, as it is not a robust characterisation. Why was mass spectrometry not in their characterisation?

Figure 2 would be much clearer if the relevant wavelengths were highlighted, perhaps with a dotted line, so that the reader can easily interpret which matrices are suitable for typical MALDI laser wavelengths (337nm (nitrogen), 345nm (Nd:YLF) and 355nm (Nd:YAG) should be highlighted). Also, the x axis label has a typo (wavelength, rather than wavelength). Furthermore, no recommendations are made as to which of the compounds might be best suited to each laser. This analysis is essential.

Much more discussion is required around Figure 3. Which compounds are the most promising in terms of lack of interfering peaks? Which in terms of highest ion counts? Why is the mass range above 700 not shown? This would be important for phospholipid analysis. Also, why is the region 50-100 not considered? If, as the authors claim, these compounds are being considered for metabolite and lipid analysis, both of these regions should be considered and discussed.

Experiments to assess use as a MALDI matrix by analysing a MALDI matrix are unusual. I rather think it would be more logical to assess the synthesised compounds in the presence and absence of a proton donor such as TFA.

Application of potential matrices to standard lipid mixtures: The properties of PE, PG and PS lipids are quite varied, and in some cases negative ion mode rather than positive ion mode presented would be more applicable. I am pleased to see that the authors have analysed lipids in both polarities, however the negative ion mode data needs to be discussed in detail. In particular, which adducts were evaluated? What are the ion counts? What degree of fragmentation was observed? Furthermore, all experiments should be repeated for individual lipids with ach matrix, and then recommendations made for each lipid type. Then they should go on to look at lipid mixtures. Which matrices led to the most significant interference peaks in terms of fragmentation of the lipids studied?

How were the samples in Figure 5 deposited? Dried droplet? Overlayer? By a spraying method? This should be clear in the figure legend. Also, a discussion of crystallisation differences between spotting and spraying methods for separate recommendations as to which compounds would be suitable for spot analysis and imaging applications would be extremely beneficial.

Which adduct or adducts are represented in Figure 6?

Application of selected matrices to real samples: Why were only certain matrices used to assess the analysis of certain complex mixtures? The rationale needs to be clearly explained. Why is plasma different?

Conclusions: A number of these are new, and not commented on earlier in the manuscript. Shot-to-shot variability should be discussed in the main text for each matrix compound. I disagree that the data presented supports the claim here that ‘all matrices exhibited good UV-absorption at the laser wavelength of 345 nm’. Rather, compounds #3, #8 and #10. Recommendations for neutral lipids specifically, and why this is the case, should be discussed in detail in the main text.

Author Response

Referee 1

Comments and Suggestions for Authors

The submitted article “Synthesis and investigation of alternative ‘second generation’ matrices for matrix-assisted laser desorption/ionization mass spectrometry analysis of lipids” describes a new suite of rationally designed MALDI matrices for improved analysis of lipids from complex samples. The authors present four new MALDI matrix compounds for improved ionisation, S/N ratios, reproducibility and reduced interference at low m/z. Different recommendations are made for neutral DAGs and TAGs and charged phospholipids.

Although this study is well presented and is informative study, unfortunately, it is my opinion that the experiments are somewhat lacking in key areas of the study, the results are not discussed as fully as they can and should be, and some of the conclusions are not fully supported. I offer the following comments:

First, we would like to thank the reviewer for his/her precious suggestions and comments.

Introduction: most of the key literature seems to be presented, although the authors self-cite rather too much.

The authors have worked on this topic for more than fifteen years; however, some self-citations have been removed.

Page 3 line 114 ‘fifthy’, rather ‘fifth’.

Done.

Results: Synthesis and characterization of novel matrices: the authors have not convincingly characterised the synthesised compounds. Table 1 presents LogP predicted by ChemDraw, theoretically calculated proton affinities and an estimated molar extinction coefficient (based on what exactly?). The only characterisation that seems to be presented is proton NMR, this is not enough and in my opinion is not acceptable, as it is not a robust characterisation. Why was mass spectrometry not in their characterisation?

The molar extinction coefficients were calculated by UV-Vis spectra while the complete routinary characterization of new compounds was reported in the Supplementary Material including carbon and proton NMR alongside accurate mass determination performed by ESI-IT-ToF-MS. Moreover, LDI MS for characterization was also employed. As suggested, such a section, previously reported in Supplementary Material, has been moved to the main text of the revised version.

Figure 2 would be much clearer if the relevant wavelengths were highlighted, perhaps with a dotted line, so that the reader can easily interpret which matrices are suitable for typical MALDI laser wavelengths (337nm (nitrogen), 345nm (Nd:YLF) and 355nm (Nd:YAG) should be highlighted). Also, the x axis label has a typo (wavelength, rather than wavelength). Furthermore, no recommendations are made as to which of the compounds might be best suited to each laser. This analysis is essential.

Following the referee’s suggestion, the typical MALDI laser wavelengths (337 nm (nitrogen), 345 nm (Nd:YLF) and 355 nm (Nd:YAG) have been inserted in Figure 2, thus making easier their comparison. The recommendations cannot be done only based on UV absorption because several parameters need to be considered. Except for compounds 3, 8, and 10, it is evident from Table 1 that each matrix exhibits a good molar extinction coefficient.

Much more discussion is required around Figure 3. Which compounds are the most promising in terms of lack of interfering peaks? Which in terms of highest ion counts? Why is the mass range above 700 not shown? This would be important for phospholipid analysis. Also, why is the region 50-100 not considered? If, as the authors claim, these compounds are being considered for metabolite and lipid analysis, both of these regions should be considered and discussed.

The discussion on the most promising matrices in terms of interfering peaks and ion counts has been extended in the revised version. Although spectra were acquired from 100 to 1000 m/z, a zoomed portion was reported both for better viewing and because no signal was found above m/z 700. However, the figure has been amended in the revised version since, as the referee suggested, it could be useful to check the region of PL and lipid above m/z 700. The very low mass region is out of interest for the present study.

Experiments to assess use as a MALDI matrix by analysing a MALDI matrix are unusual. I rather think it would be more logical to assess the synthesised compounds in the presence and absence of a proton donor such as TFA.

The experiments were performed according to Jaskolla et al. in PNAS 2008, 105, 12200, in which CHCA as a reference matrix with a known proton affinity was used to rationalize the proton transfer based on their difference in PA. Indeed, the higher acidity of a protonated adduct or its lower PA leads to an efficient proton transfer to CHCA, and thus it could be assumed a more efficient proton transfer to analytes.

The experiment suggested by the Referee has been employed by Shroff, R., Svatoš in PNAS 2009, 106, 10092, to explain a completely different mechanism in solid solutions using a strong base (proton sponge) as a novel matrix for MALDI-TOF/MS analysis of anions. By applying the Brønsted–Lowry model, the authors speculate an equilibrium between the ion pair and the associated base–acid complex in the solution which is characterized by reciprocal pKa. According to this suggested approach, the ion yield should depend on both the matrix/analyte ratio and the pKa, so it could be reasonable to check the behavior of the matrix mixed with acids of different strengths.

Application of potential matrices to standard lipid mixtures: The properties of PE, PG and PS lipids are quite varied, and in some cases negative ion mode rather than positive ion mode presented would be more applicable. I am pleased to see that the authors have analysed lipids in both polarities, however the negative ion mode data needs to be discussed in detail. In particular, which adducts were evaluated? What are the ion counts? What degree of fragmentation was observed? Furthermore, all experiments should be repeated for individual lipids with ach matrix, and then recommendations made for each lipid type. Then they should go on to look at lipid mixtures. Which matrices led to the most significant interference peaks in terms of fragmentation of the lipids studied?

We are aware that lipids can be analyzed in negative ion mode, especially some lipid classes such as acidic PI and PG, also including PE and PS, that could be detected as deprotonated molecules by high-performing matrices such as 9-aminoacridine, protons sponges, DAN, norharmane, etc. We know that PC and SM can be detected in negative ion mode as demethylated species. Nevertheless, when lipids were investigated in negative mode, the proposed matrices do not work well as does the archetype CHCA. For this reason, we did not further investigate these compounds in negative ion mode.

How were the samples in Figure 5 deposited? Dried droplet? Overlayer? By a spraying method? This should be clear in the figure legend. Also, a discussion of crystallisation differences between spotting and spraying methods for separate recommendations as to which compounds would be suitable for spot analysis and imaging applications would be extremely beneficial.

The samples were deposited by using the dried-droplet method that still provides the “gold standard” for analyses by MALDI-MS. This information has been added to the revised manuscript. We aimed to compare the compounds considering PA, molar extinction coefficients, and crystallization behavior. This latter does not discriminate, and its effect was unpredictable being strongly influenced by small changes in the ambient moisture, temperature, roughness of the target, and solvent used. Many experimental parameters should be considered to rationalize its effect.

Which adduct or adducts are represented in Figure 6?

The description of the adducts considered has been inserted in the caption of Figure 6.

Figure 6. Heatmap plots in logarithmic scale showing the normalized average intensity of each compound mixed with each matrix in positive (upper panel) and negative ion mode (lower panel). In positive ion mode, the intensities are the sum of sodium and protonated adducts. For TAG, DAG, MAG, PG, and PI the protonated adduct was not observed regardless of the matrix used while the sodium adduct was observed for all lipids. In negative ion mode, the deprotonated molecules were considered for PE, LPE, PG, PI, PS and Cer while the demethylated species for PC, LPC, and SM. TAG, DAG, MAG and CholEst do not ionize in negative modality. The color scale index goes from the highest (yellow) to the lowest (blue) intensity. “

Application of selected matrices to real samples: Why were only certain matrices used to assess the analysis of certain complex mixtures? The rationale needs to be clearly explained. Why is plasma different?

The experimental work on lipid standards alone and in the mixture suggested to us that the chosen matrices were the most performing for complex real samples. Of course, all matrices were tested with a milk extract, but the best significant results were obtained in only a few of them. Then, the selected matrices 1, 2, 6, and 7 were applied to investigate complex lipids in extracts of cow milk, fish, human plasma, and blood; figure 7 reports just some representative spectra but the matrices genereally worked with all the examined samples.

Conclusions: A number of these are new, and not commented on earlier in the manuscript. Shot-to-shot variability should be discussed in the main text for each matrix compound. I disagree that the data presented supports the claim here that ‘all matrices exhibited good UV-absorption at the laser wavelength of 345 nm’. Rather, compounds #3, #8 and #10.

A discussion on the spot-to-spot variability was included in the revised version of the manuscript. As some matrices did not give satisfactory results, in the conclusion section the sentence “all the matrices” was properly amended with “Seven matrices...”.

“To compare the response of all compounds in the PL analysis, the intensities of a double sodiated adduct [DMPG+2Na-H]+ were registered in triplicate on a total of 100 spectra per spot and spot-to-spot. The reproducibility was estimated as relative standard deviation (RSD). The evidence of the higher degree of homogeneity for compounds 1, 2, 4, 5, 6, 7 was obtained already during the spectra acquisition; the occurrence of a [DMPG+2Na-H]+ signal was always detected in each sampling point whereas several ‘cold’ points of very low intensity were observed with matrices 3, 8, 9, and 10. Accordingly, the spot-to-spot reproducibility as RSD was respectively 15%, 18%, 20%, 15%, 22%, 21% for matrices 1, 2, 4, 5, 6, 7, and 35%, 45%, and 55% for matrices 10, 3, and 8, following a dried-droplet deposition.

Recommendations for neutral lipids specifically, and why this is the case, should be discussed in detail in the main text.

As suggested, the following sentences were added.

“In the case of matrix 2, it is argued that, due to the strong detection of TAG and DAG sodium adducts, not only the PA but also the sodium cation affinity (SCA) play a pivotal role. Jaskolla et al. [57] rationalize the improved performance of CClCA in detecting the phosphatidylethanolamine chloramines as cation adduct by advoking a lower SCA compared to CHCA (i.e. 177.9 kJ/mol vs 192.5 kJ/mol). They explained that the stable conformation of both CHCA-Na and CClCA-Na complexes is due to sodium chelation of the carbonyl oxygen of the acid functionality and the nitrogen atom. Being the PA value of compound 2 close to CClCA (802.3 kJ/mol vs 804.6 kJ/mol) we suggested a similar behavior for SCA due to the occurrence of strong electron withdrawal groups, i.e. nitro and chloro substituent, respectively, which can lead to a lower stabilization of the sodiated complex compared to the other matrices.”

Reviewer 2 Report

The article by Monopoli  et al., titled “Synthesis and investigation of alternative ‘second generation’ matrices for matrix-assisted laser desorption/ionization mass spectrometry analysis of lipids” reports the synthesis of new acid derivatives as matrices for MALDI studies of Lipid samples, their UV study and use as a matrices in MALDI of lipids. The current form of the manuscript cannot be accepted. However, if the authors rewrite it with answers to following point it can be considered.

The introduction; line 48-49 “From a general … there exists”. The sentence would be fine without there. The English language and style need major correction, it’s too weak and full of errors (grammatical and spelling). From scientific perspective the reason for why authors didn’t find MALDI matrix is because, it is not necessary for lipid studies or lipidomics. ESI-MS, LC-MS, UPLC-MS and so forth have been working very well. Thus, authors need to do a better introduction and justify the need of their study properly.

Line 51 as wide as… line 114 fifthy?... please correct these kinds of phrases throughout the manuscript. There are many more. Please don’t use redundant words or phrases.

Interestingly, at three places the authors chose not to show the data and mention in italic. There is no limit in SI material, I suggest the authors to provide everything in SI.

Further, in the chemical synthesis of matrices, the authors have referred to (Figures 3-12), this should be changes to S3-12. On looking inside SI, one finds S.2 as general procedure of…. Of what? Matrices. I guess a better way is to mention it. Then it mentions to look down (vide infra). And then the NMR spectra and data is found from page 5 of SI. This would be too inconvenient to the readers to dig for data in the manuscript and SI. I would suggest to draw a chemdraw scheme illustrating synthesis in the current figure 1 of the manuscript. And put these NMR data under the chemical synthesis of matrices part in the text. The SI can have the NMR spectra images only.

The authors have used hash for numbering compound, #1, #2,.. It’s wrong. The number 1, 2 or any number, or bold numerals like 1 would be the right way.  

The synthetic strategy is based on routine chemistry. And the protocols given for recrystallization suggests that the compounds were precipitated and not crystallized. Did the authors obtain crystals? Are any of the chemical structure new? If yes, they should obtain a CCDC no by depositing the structure to Cambridge Crystallographic Data Centre.

As shown by the figure, MS spectra many of the matrices are not performing well the S/N ratio is quite big.

Are any of the SO-CALLED first-generation matrices marketed? Why was a comparison data not shown?

The authors should change the title to “Synthesis and investigation of new matrices for matrix-assisted laser desorption/ionization mass spectrometry analysis of lipids”. The current study and manuscript do not have sufficient study for using term second generation, which is not acceptable.

Author Response

Referee 2

The article by Monopoli et al., titled “Synthesis and investigation of alternative ‘second generation’ matrices for matrix-assisted laser desorption/ionization mass spectrometry analysis of lipids” reports the synthesis of new acid derivatives as matrices for MALDI studies of Lipid samples, their UV study and use as a matrices in MALDI of lipids. The current form of the manuscript cannot be accepted. However, if the authors rewrite it with answers to following point it can be considered.

The introduction; line 48-49 “From a general … there exists”. The sentence would be fine without there. The English language and style need major correction, it’s too weak and full of errors (grammatical and spelling).

We thank the referee for her/his comments. The manuscript has been amended in grammatical and spelling errors.

From scientific perspective the reason for why authors didn’t find MALDI matrix is because, it is not necessary for lipid studies or lipidomics. ESI-MS, LC-MS, UPLC-MS and so forth have been working very well. Thus, authors need to do a better introduction and justify the need of their study properly.

Although we are fully aware of lipidomics studies by LC-ESI-MS that our group has largely employed, MALDI-MS has still a lot of advantages for lipid analysis. This viewpoint has been more emphasized in the revised manuscript.

Line 51 as wide as… line 114 fifthy?... please correct these kinds of phrases throughout the manuscript. There are many more. Please don’t use redundant words or phrases.

Done.

Interestingly, at three places the authors chose not to show the data and mention in italic. There is no limit in SI material, I suggest the authors to provide everything in SI.

The points where we specified data not shown were somehow pleonastic and this specification was superfluous in many cases since the data and results were included in other forms in the text. For instance, a comparison between CHCA and CClCA was performed, and the results are included in Figure S2. Even if the SI has no limits, we would like to limit the number of figures and materials strictly functional to the work.

Further, in the chemical synthesis of matrices, the authors have referred to (Figures 3-12), this should be changes to S3-12.

Correction done.

On looking inside SI, one finds S.2 as general procedure of…. Of what? Matrices. I guess a better way is to mention it.

According to the reviewer’s suggestion, we have moved these general remarks in the text and deleted them from SI.

Then it mentions to look down (vide infra). And then the NMR spectra and data is found from page 5 of SI. This would be too inconvenient to the readers to dig for data in the manuscript and SI. I would suggest to draw a chemdraw scheme illustrating synthesis in the current figure 1 of the manuscript. And put these NMR data under the chemical synthesis of matrices part in the text. The SI can have the NMR spectra images only.

As described in the text, matrices were obtained by standard Knoevenagel condensation reactions. Synthetic details are given in the S.I., including the complete routinary characterization together with the chemical shift of NMR data and accurate mass values.

The authors have used hash for numbering compound, #1, #2,.. It’s wrong. The number 1, 2 or any number, or bold numerals like 1 would be the right way.

We have decided to insert hashtags #1, #2, to facilitate the compounds’ recognition and to distinguish them from other numbers such as the m/z values shown in the text. However, following the referee’s suggestion, we have changed the style by using bold numerals.

The synthetic strategy is based on routine chemistry. And the protocols given for recrystallization suggests that the compounds were precipitated and not crystallized. Did the authors obtain crystals? Are any of the chemical structure new? If yes, they should obtain a CCDC no by depositing the structure to Cambridge Crystallographic Data Centre.

On this point, the reviewer is right since the compounds were reprecipitated and not recrystallized. Accordingly, the manuscript has been properly amended. As detailed in the text moreover, all new compounds have been fully characterized by routinary NMR (1H and 13C) and accurate m/z values. UV and LDI MS analyses were also performed. The X-rays data are missing since they are out of the scope of the present manuscript. We do believe that these data do not add significant information at the present work.

As shown by the figure, MS spectra many of the matrices are not performing well the S/N ratio is quite big.

Sorry, we do not get the point since a high S/N ratio means a well-performing matrix. We reported the results in terms of S/N in Figure S2; for matrices 3 and 10 we generally obtained low S/N ratios.

Are any of the SO-CALLED first-generation matrices marketed? Why was comparison data not shown?

The so-called first-generation matrices are CHCA, DHB, sinapinic acid, etc., i.e. matrices that were empirically chosen in the early days of MALDI-MS after testing hundreds of compounds. We have performed a comparison with the archetype CHCA and the well-known second-generation matrix CClCA that is nowadays commercially available as shown in Figures 5 and S2.

The authors should change the title to “Synthesis and investigation of new matrices for matrix-assisted laser desorption/ionization mass spectrometry analysis of lipids”. The current study and manuscript do not have sufficient study for using the term second generation, which is not acceptable.

We do not agree with the reviewer on this point since, as clarified by Bahr and Jaskolla (Employing ‘second generation’ matrices. In Advances in MALDI and Laser-Induced Soft Ionization Mass Spectrometry; Springer International Publishing, 2015; pp. 3–35 ISBN 9783319048192.), the second-generation matrices are obtained by modifying the structures of established matrix by changing the nature, number, positions of functional groups as we did in this manuscript by using CHCA as the well-established starting matrix.

In comparison with the well-known second-generation matrix ClCCA, some of the proposed matrices are superior in terms of S/N ratio and reproducibility as proved in the present manuscript for lipids analyses.

Reviewer 3 Report

In this paper, ten different CHCA derivatives were prepared and analyzed as MALDI matrices for lipid analysis. The authors evaluated the crystallographic properties and mass spectrometry patterns of different matrices and applied them to the lipid analysis of real samples. The results showed that matrix #7 was screened as the best matrix for PL analysis, and matrix #2 was better for the analysis of neutral lipids such as DAG and TAG. This paper develops new mechanism based on CHCA and CICCA, which is innovative and effective. I think this manuscript can be published on "Molecules". However, we have some suggestions for the authors to check and answer. The specific suggestions are as follows.

  1. Positive MALDI-ToF MS spectra of the complex lipid extracts from cow milk, fish, plasma, and blood was described in detail. We would like to ask if the author has used these new matrix for imaging of actual slice samples, such as fish slices, etc. It would be more intuitive if there were some imaging pictures.

  1. In figure 5, “CICCA” is not clearly displayed.

  1. Line 274 has a full stop in the title, suggest checking and modifying it.

  1. The description of the effect of matrix evaluation in this paper is vague. For example, line 321, "matrix #1 gave rise to a simpler spectrum with a higher S/N ratio, whereas matrix #6", does not have some precise multiplicative relationship to judge characteristics.

  1. In line 341, a clear division between experimental methods (materials, chemical synthesis of matrices, etc) and “results and discussion” is recommended.

6. The conclusion is the only capitalized title, and we suggest unification.

Author Response

Reviewer 3

In this paper, ten different CHCA derivatives were prepared and analyzed as MALDI matrices for lipid analysis. The authors evaluated the crystallographic properties and mass spectrometry patterns of different matrices and applied them to the lipid analysis of real samples. The results showed that matrix #7 was screened as the best matrix for PL analysis, and matrix #2 was better for the analysis of neutral lipids such as DAG and TAG. This paper develops new mechanism based on CHCA and CICCA, which is innovative and effective. I think this manuscript can be published on "Molecules".

We thank the referee for commenting on the value of our manuscript.

However, we have some suggestions for the authors to check and answer. The specific suggestions are as follows. Positive MALDI-ToF MS spectra of the complex lipid extracts from cow milk, fish, plasma, and blood was described in detail. We would like to ask if the author has used these new matrices for imaging of actual slice samples, such as fish slices, etc. It would be more intuitive if there were some imaging pictures.

We have analyzed only lipids isolated after liquid/liquid or liquid/solid extraction following the Bligh Dyer protocol. Unfortunately, we cannot perform imaging analysis, but we are confident that these compounds could work also for slice samples.

In figure 5, “CICCA” is not clearly displayed.

Correction done.

Line 274 has a full stop in the title, suggest checking and modifying it.

Correction done.

The description of the effect of matrix evaluation in this paper is vague. For example, line 321, "matrix #1 gave rise to a simpler spectrum with a higher S/N ratio, whereas matrix #6", does not have some precise multiplicative relationship to judge characteristics.

A more detailed description has been inserted.

In line 341, a clear division between experimental methods (materials, chemical synthesis of matrices, etc) and “results and discussion” is recommended.

The title of the Experimental Section has been inserted.

  1. The conclusion is the only capitalized title, and we suggest unification.

Done.

Round 2

Reviewer 1 Report

Thank you for your consideration of my comments, I am pleased with the revised manuscript.

Author Response

We thank the reviewer for his/her positive comments.

Reviewer 2 Report

The current version of the manuscript is somewhat improved than the first draft. However, despite the opportunity to make it publishable they have completely disregarded some of the comments. The major one being their language is very weak. They had the opportunity to rewrite the manuscript, but it is still full of basic grammatical errors, the usage of singular plural etc. To list a few, right from the beginning of the manuscript, the use of phrases, Line 40, remain as, for; Line 45, it has experimented; Line 81 de-finite; Line 89 with as low as possible is wrong and redundant. The way it is written, the language in whole manuscript is pleonastic and the specifications are superfluous.

The title is NOT acceptable. The term second generation is borrowed from a chapter in a book published by Springer. The current work includes a few modifications and synthesis of some derivative molecules, which were used as MALDI matrix. They are not the inventors of second generation matrix. To quote ‘second generation’ in title is highlighting a term or study. Even if they want to include the term, they have to do away with the quote and write it as a running title.

Author Response

We thank the reviewer for his/her positive comments. We have changed the title and revised the whole manuscript.

Reviewer 3 Report

The manuscript has been significantly improved by revisions and can be accepted after grammatical and technical checks.

Author Response

We thank the reviewer for his/her positive comments. We have checked for grammar errors.